# Selection and Evaluation of 21 Potato (*Solanum Tuberosum*) Breeding Clones for Cold Chip Processing

**DOI:** 10.3390/foods8030098

**Published:** 2019-03-14

**Authors:** Benjamin Opuko Wayumba, Hyung Sic Choi, Lim Young Seok

**Affiliations:** 1Department of Bio-Health Technology, Kangwon National University, Chuncheon 200-701, Korea; bwayumba@gmail.com (B.O.W.); luckychs@naver.com (H.S.C.); 2KPBR (Korea Potato Breeding Resource Bank), Kangwon National University, Chuncheon 200-701, Korea

**Keywords:** chip processing, cold storage, reconditioning, reducing sugar, potato

## Abstract

Quality evaluations in potatoes are of necessity to meet the strict demands of the chip processing industry. Important parameters assessed include specific gravity, dry matter content, chip color, reducing sugars, and glycoalkaloids. This study was designed with the purpose of identifying specialized potato clones with acceptable qualities for processing chips, in comparison with two selected control varieties, Dubaek and Superior. As a result, high dry matter and specific gravity were observed for three potato clones, and the quantified ά-solanine levels ranged from 0.15 to 15.54 mg·100 g^−1^ fresh weight (FW). Significant variations (*p* < 0.05) in reducing sugar levels were observed in clones stored at different temperature conditions. After reconditioning of the tubers at 22 °C for 21 days, a significant drop in reducing sugar levels was recorded. In addition, fried chips for each potato clone were evaluated, and the color measured on the basis of the Snack Food Association (SFA) chip color score standard. Reconditioned tubers exhibited much lighter and better chip color compared to their counterparts cold-stored at 4 °C. This study observed that for quality processing of potato chips, clones with combined traits of high dry matter, low levels of glycoalkaloids and reducing sugars, and acceptable chip color should be used as raw materials.

## 1. Introduction

The importance of potatoes as food includes their use as processed products, since they can be processed into many value-added food items like chips (crisps), dried flakes, French fries, and various snacks. Processing of potatoes into chips and other products has a great potential in ensuring reduced loss and waste post-harvesting, in handling and storage.

The industry requirements for processing chips in terms of tuber appearance should be of shallow eyes, appropriate weight, and round-oval shape [1], whereas long-oval-shaped tubers are preferred for processing French fries. In addition, tubers should be free from cracks, hollow heart, secondary damage, rusty spots, and greening. The peel color, flesh color, and flour content should also satisfy the national consumer preference of any given country.

In the recent years, there has been increasing demand for food convenience, and potato chips meet these requirements. Potato chips are one of the most convenient ways to serve potatoes, with minimal preparation needed. Potato chips can be made flavored, plain, with chili, cheese, or seasoning. Their slicing can be plain, regular, wavy, or waffle-cut. They are a very popular food item for picnics and parties and can be served conveniently any time as a snack. This withstanding, the quality variation of the raw materials used to produce chips has led to large differences in their culinary value and consumer acceptance. Therefore, it is important to address this issue prior to processing in order to maintain taste and flavor. Potato flavor results from the combination of taste, aroma, and texture [2]. However, the resultant flavor after processing is sometimes altered because of the effects of cold storage. Although the storage of potato tubers at low temperature (4 °C) minimizes tuber respiration and sprouting [3], low-temperature storage also activates a process known as low-temperature sweetening that results in the conversion of starch to reducing sugars. High levels of reducing sugars (glucose and fructose) lead to undesirable non-enzymatic browning reactions and the formation of amine groups of free amino acids [4] that cause off-flavors and darkening of the processed potato products [5] during frying.

In order to mitigate this problem, it is necessary to develop and identify cultivars with first-rate qualities that can be a game changer in the chip processing industry. In this regard, this study was designed with the purpose of identifying specialized potato clones with acceptable tuber qualities for processing chips with respect to tuber specific gravity, dry matter content, chip frying tests, solanine content, and reducing sugar profiles under different temperature storage regimes.

## 2. Materials and Methods

### 2.1. Plant Materials

Among potato breeding clones grown in 2017 spring season in South Korea, Gangwon-do Province, 21 breeding clones and 2 control varieties (Dubaek & Superior) were finally selected for further evaluation for chip processing. A criterion of fully mature and healthy tubers of high yield were used to select the tubers. The selected tubers from each clone were stored at different temperatures: 22 °C at harvest, 4 °C during cold storage, and 10 °C for 3 weeks before experimental analysis. The cold-stored tubers were then reconditioned at 22 °C for additional three weeks and re-evaluated.

### 2.2. Dry Matter (DM) Content

Clean, dry, standard-size aluminum foil boats were used as crucibles. Sliced samples, 3 mm thin, (10 g) of each potato clone were weighed in the foil boat, and the initial weight measured in grams. The samples were dried in an electric oven overnight for 16 h at 105 °C to a constant weight. The total solid content of each clone was calculated as a percentage. Four 10 g samples were measured for each clone. (DM % = (final dried weight/initial weight) × 100).

### 2.3. Specific Gravity (SG)

Six (80–130 g) tubers from each potato clone were weighed in air and under water. Tubers were all weight-matched to ensure uniformity per clone. Average underwater weights were used to calculate the specific gravity.SG = (weight in air/ weight in water × density water (g·cm^−3^))

### 2.4. Reducing Sugar

The reducing sugar was quantified by the dinitrosalicylic acid (DNS) method [6]. For the procedure, 1 g of freeze-dried powdered sample was added to 50 mL of distilled water and shaken for 1 h. The supernatant was centrifuged at 3000 rpm for 30 min. In the experiment, 2 mL of DNS was added to 1 mL of sample in a glass tube (15 × 100) and incubated at 99 °C in a water bath for 10 min. After cooling, the absorbance was measured at 550 nm in a microplate reader. The experiment was replicated twice. The samples were analyzed against glucose standards of known concentrations.

### 2.5. Chip Frying Test

Potato tubers of different clones were washed clean and sliced from apical to basal ends. Thin slices (1 mm) were cut using a hand-held slicer. The slices were then rinsed in water to remove excess starch and blot-dried on paper towels. A deep-fryer machine containing vegetable oil was used to prepare the chips. At a constant temperature of 180 °C, the chips were fried for 2 min and then placed on a paper towel to drain off excess oil. The chips were then photographed after cooling, using a high-resolution Canon EOS 5D Digital SLR Camera (Chuncheon, Korea), and their color measured on the basis of the Snack Food Association (SFA) chip color measurement standard.

### 2.6. Glycoalkaloid Analysis

Tubers of 21 clones and 2 control varieties were used. The samples were sliced whole with the skin intact and freeze-dried (ilshin Lab Co., LTD, Chuncheon, Korea) prior to glycoalkaloid content analysis to determine ά-solanine. All reagents for the analysis were prepared in HPLC-grade deionized water. The ά-solanine standard and acetonitrile HPLC-grade were purchased form Sigma Aldrich (Chuncheon, Korea). All other chemicals used were of standard analytical grade.

Analysis was carried out according to Hellenas et al., 1995a [7], with some modifications. In the experiment, 10 g powder extracted from a whole tuber was mixed with 20 mL water/acetic acid/sodium bisulphate (N_a_HSO_3_) in the ratio 95:5:0.5 (v/v/w) in a blender. The mixture was diluted up to 50 mL using the same extraction solvent and vacuum-filtered through Whatman filter paper No.1. The filtrate was further cleaned up by centrifugation at 6500 rpm for 10 min. A volume of 5 mL of supernatant was obtained which was further cleaned up by extraction with 1mL acetonitrile, 1 mL water/acetic acid/N_a_HSO_3_ solvent, 0.8 mL water/acetonitrile, 0.8 mL acetonitrile/0.022 M potassium phosphate buffer, pH 7.6, 55:45 (v/v), all preconditioned and filtered.

ά-solanine was quantified using High-performance liquid chromatography (HPLC) apparatus (Shimadzu Corp, Tokyo, Japan) consisting of a 250 × 4.6 mm NH_2_ analytical column; mobile phase of acetonitrile/0.022 M potassium dihydrogenphosphate (KH_2_PO_4_) buffer, pH 4.7, 75:25 (v/v), at a flow rate of 1.5 mL/min, UV absorption of 200 nm wavelength, and 0.05 Absorbance units full scale (AUFS) sensitivity. The injection volume used was 20 µL. The retention time for ά-solanine was approximately 5.7 min, and sample extracts were quantified by comparing their corresponding peak areas with those of known amounts of standard. All samples were replicated twice and read against solanine standards of known concentrations.

### 2.7. Statistical Analysis

The samples’ mean variations were analyzed by analysis of variance (ANOVA) software using IBM Corp. IBM SPSS Statistics for Windows, Version 23.0 Armonk, NY, USA. Significant differences were evaluated using Tukey’s test at a 95% confidence interval.

## 3. Results and Discussion

### 3.1. Dry Matter Content and Specific Gravity

The potato clones varied with respect to dry matter content and specific gravity (Table 1). Tuber dry matter ranged from 18.37 ± 1.08 to 25.10 ± 0.88%, and specific gravity from 1.079 ± 0.006 to 1.096 ± 0.005. For both parameters, four specific clones (V50, V48, N109-35, N357) performed better than the control variety Dubaek (DM 23.22%, SG 1.088) with *p* < 0.05, while other 11 cultivars showed a similar trend, with significantly higher total solids and specific gravity levels compared to Superior (DM 21.92%, SG 1.078).

The observed differences in dry matter and specific gravity among potato clones may be mainly due to genetic constitution, since all clones were grown and tested in one location with similar management. Tuber dry matter content is the most important attribute that determines the quality and yield of fried and dehydrated products. Higher dry matter or total solids result in higher recovery of the processed products, lower oil absorption, less energy consumption, and crispier texture [8,9].

Based on Pearson correlation test (r = 0.696) performed in the study, it was observed that clones with high specific gravity exhibited significantly (*p* < 0.01) high dry matter content, which is an important feature in the selection for processing chips. Tuber specific gravity is often used in the processing industry as a means for a quick estimation of total solids, as the two parameters are highly correlated [10,11].

### 3.2. Reducing Sugar

Significant (*p* < 0.05) variations in the reducing sugar content of the potato clones were observed under different temperature storage regimes. Overall, the reducing sugar content across the three temperature storage conditions ranged from 2.24 to 40.84 mg·g^−1^ dry weight (DW) (Table 2).

For the tubers analyzed at harvest, three clones (V63, N357, V50) exhibited significantly lower levels of total reducing sugars compared to the main control variety ‘Dubaek’, while another set of genotypes (N109-35, N189, V93, V48, V16, V18) showed just as great processing aptitude potential at harvest, with similar sugar levels as the control.

At 4 °C cold storage, the clones (V93 & V48) exhibited the desired low reducing sugar levels alongside the control variety ‘Dubaek’. Such cultivars with cold chipping properties are of great importance to the chip processing industry as they can provide high-quality material for processing all year round and thus increase industry efficiency.

At 10 °C cold storage, the overall mean reducing sugars of the genotypes were considerably low (Table 2), and many clones (N357, V17, V50, V63, V93, V48, A9) showed great aptitude for processing in terms of the desired total reducing sugar levels. This could be attributed to the fact that, in general, potato tubers for chip processing are stored at temperatures of about 8 to 12 °C in order to avoid an increase in sugar content [12].

At harvest, the potato clones’ mean levels of reducing sugars (Table 2) were lower than those of their counterparts cold-stored at 4° and 10 °C. This has also been noted by Hayes and Thill [13] and Meena et al. [14] who observed that ‘tubers have low levels of reducing sugars and produce light-colored chips directly after harvest’. The high levels of sugars in 4 °C stored tubers occurred as a result of low-temperature sweetening (LTS). During LTS, potato tubers accumulate sucrose and reducing sugars (glucose and fructose). This accumulation of sugars at low temperature causes a reduction in starch content and ultimately affects the quality of the fried products [15].

Among other parameters, the reducing sugar level is considered one of the most limiting factors in cold chip processing. The sugars in tubers react with proteins and free amino acids in the cytoplasm during the frying process, producing dark pigments with a bitter taste that devalue the final product [16]. Therefore, the importance of using material with low sugar contents is further emphasized by our study results.

### 3.3. Reconditioning Effect on Potato Clones

After cold storage at 4 °C, the potato clones were put at ambient temperature of 22 °C for 21 days in order to evaluate their reconditioning ability. During reconditioning, accumulated sugars are eliminated in the increased storage temperature. The decrease in sugar levels results from respiration as well as reconversion of sugars to starch [17].

From the results (Table 3), it was seen that two genotypes (V50, V93) that had performed relatively poorly in 4 °C cold storage were able to recondition well enough alongside the main control variety. Another set of genotypes (V48, N109-35, N2-6, A9) also showed high reconditioning ability, with the clone V48 showing the lowest and safest reconditioning ratio (Table 3) among the top performers. A genotype with safe reconditioning ratio would be desirable for the chip processing industry as it exhibits consistent reducing sugar levels both in cold storage and in reconditioned states. The reconditioning ratios were calculated and expressed as a percentage of reconditioned tubers against non-reconditioned ones.
Recondition Ratio (%) = 100 − (Reconditioned state/cold-storage state × 100)

On the basis of the reconditioning trends, we observed that not all potato cultivars reconditioned positively, and although some clones (A165-4, V17, A12-5) showed a great come-back ability, their reducing sugar levels were still not within the acceptable range. This is a confirmation that the degree to which quality can be restored with reconditioning varies with cultivar [18].

### 3.4. Chip Frying Test

A major indicator of chip quality after frying is the resultant chip color. This attribute can be adequately measured on the SFA color scale (Figure 1), with the lighter color (score 1.0) being highly desirable, and the darker color (score 5.0) considered as unacceptable. The SFA chip color scores of the fried potato clones stored at different temperature conditions were evaluated and recorded (Table 4).

For tubers fried after 4 °C cold storage, four clones (A9, V93, V48, A165-4) exhibited acceptable light chip color alongside the control ‘Dubaek’. Among these, only two clones (V93, V48) showed consistency in chip color and their corresponding reducing sugar content (Table 2, Figure 2).

The study observed that reconditioning tubers at 22 °C for 21 days had a positive effect on many clones (A9, V93, N357, V48, A165-4, V50, N2-6, N69-2) alongside the experimental controls Dubaek and Superior. Among these, five clones (A9, V93, V48, V50, N2-6) were graded as acceptable on the basis of the corresponding light chip color and low reducing sugar levels (Table 2, Figure 3).

### 3.5. Glycoalkaloids

The isolation of glycoalkaloids from tubers is based on the fact that the major alkaloids are soluble in slightly acidified water [19]. The glycoalkaloid (ά-solanine) contents in tubers showed wide variation among the potato genotypes, ranging from 0.15 to 15.54 mg·100 g^−1^ fresh weight (FW). In the study, one cultivar (V50) exhibited undetectable levels of ά-solanine alongside the control variety, and although only one of the major glycoalkaloids (ά-solanine) was quantified, the levels were generally considered acceptable with regards to potato chip processing, on the basis that the total glycoalkaloid content considered safe for consumption should not exceed 20 mg·100 g^−1^ FW [20]. The variations observed in solanine contents (Figure 4) could be attributed to genetically inherited differences of the potato clones and, on a broader perspective, to environmental factors during growth and storage. 

While thermal processing of tubers such as boiling potatoes in water and frying partly eliminate glycoalkaloids, processes such as baking, rinsing, cooking, slicing, and pulsed electric field have no significant effect on glycoalkaloid reduction [21]. Therefore, it is of utmost importance in processing chains that glycoalkaloid levels be kept low, especially when introducing new cultivars in the market.

## 4. Conclusions

The reconditioning of potato clones at 22 °C for up to 21 days was seen to be effective for eight potato clones. We observed that for quality processing of potato chips, clones that possess combined traits of high dry matter, low levels of reducing sugars, and low levels of glycoalkaloids with acceptable light chip color should be used as raw materials. On the basis of these parameters, we identified three clones (V93, V48, A9) that exhibited an outstanding chip processing aptitude both in cold storage and in reconditioned states. These clones were finally selected for recommendation to the chip processing industry in Korea and beyond.

## Figures and Tables

**Figure 1 foods-08-00098-f001:**
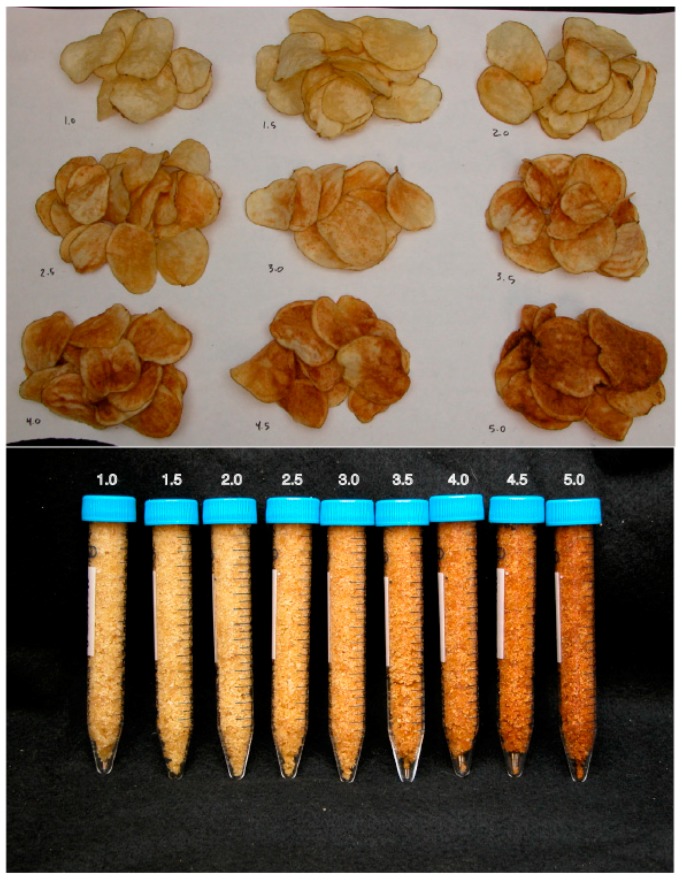
Snack Food Association (SFA) chip color measurement standard.

**Figure 2 foods-08-00098-f002:**
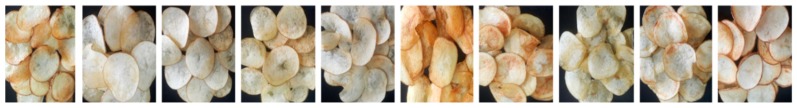
Fry chip color of samples prepared from clones cold-stored at 4 °C: V50, V93, Dubeak, V48, Superior, N109-35, N2-6, A9, A165-4 and V17 from left to right respectively.

**Figure 3 foods-08-00098-f003:**
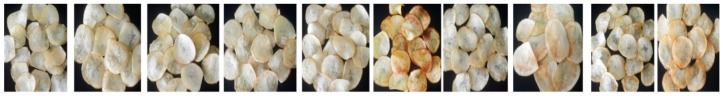
Fry chip color of clone samples reconditioned at 22 °C: V50, V93, Dubeak, V48, Superior, N109-35, N2-6, A9, A165-4 and V17, from left to right.

**Figure 4 foods-08-00098-f004:**
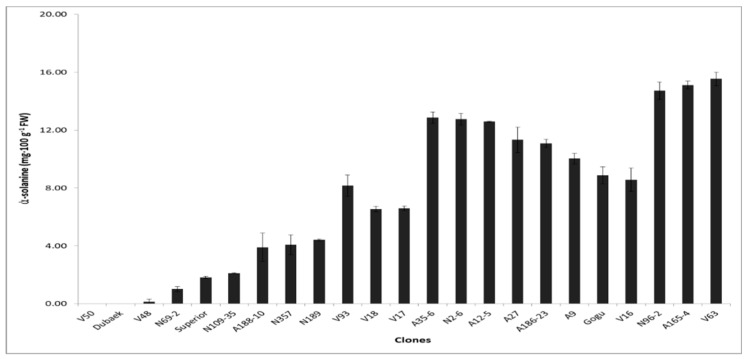
Solanine content in potato clones. Significant mean differences shown with *p* < 0.05 in Tukey’s test.

**Table 1 foods-08-00098-t001:** Tuber dry matter content and specific gravity of potato clones.

Clones	Dry Matter (%)	Specific Gravity
V50	25.10 ± 0.88 ^a^	1.096 ± 0.005 ^a^
V48	24.84 ± 0.30 ^a^	1.093 ± 0.008 ^ab^
N109-35	24.75 ± 0.63 ^ab^	1.092 ± 0.006 ^abc^
N357	24.02 ± 0.42 ^abc^	1.091 ± 0.005 ^a–d^
Dubaek	23.22 ± 0.41 ^a–d^	1.088 ± 0.006 ^a–e^
V93	23.14 ± 0.51 ^a–d^	1.082 ± 0.005 ^a–g^
A165-4	23.06 ± 0.74 ^a–d^	1.076 ± 0.010 ^d–g^
N2-6	22.77 ± 0.44 ^a–d^	1.086 ± 0.010 ^a–f^
N96-29	22.43 ± 1.19 ^b–e^	1.074 ± 0.008 ^efg^
Gogu	22.36 ± 0.57 ^cde^	1.073 ± 0.003 ^fg^
V18	22.27 ± 0.25 ^cde^	1.079 ± 0.008 ^b–g^
N189	22.26 ± 0.72 ^cde^	1.079 ± 0.004 ^b–g^
V17	22.22 ± 1.60 ^cde^	1.086 ± 0.008 ^a–f^
V16	21.97 ± 0.85 ^cde^	1.081 ± 0.004 ^b–g^
A188-10	21.84 ± 0.74 ^cde^	1.069 ± 0.008 ^g^
A9	21.77 ± 0.24 ^cde^	1.082 ± 0.007 ^a–g^
Superior	21.92 ± 0.69 ^cde^	1.078 ± 0.005 ^c–g^
N69-2	21.67 ± 2.50 ^cde^	1.083 ± 0.008 ^a–g^
V63	21.64 ± 0.36 ^cde^	1.082 ± 0.008 ^a–g^
A27	21.60 ± 0.74 ^de^	1.080 ± 0.009 ^b–g^
A186-23	20.33 ± 0.51 ^ef^	1.074 ± 0.006 ^efg^
A12-5	20.20 ± 0.93 ^ef^	1.071 ± 0.005 ^g^
A35-6	18.37 ± 1.08 ^g^	1.079 ± 0.006 ^b–g^

Table results expressed as mean ± standard deviation. The means with different letters in each column are significantly different with *p* < 0.05 in Tukey’s test.

**Table 2 foods-08-00098-t002:** Tuber reducing sugar contents in different cold storage conditions.

Reducing Sugar (mg·g^−1^ dry weight (DW))			
Clones	At Harvest (22 °C)	Cold Storage (4 °C)	Cold Storage (10 °C)
V50	2.43 ± 0.07 ^a^	7.02 ± 0.10 ^c^	2.64 ± 0.01 ^ab^
V93	2.69 ± 0.04 ^ab^	3.43 ± 0.04 ^b^	2.83 ± 0.04 ^abc^
Dubaek	2.77 ± 0.03 ^ab^	2.56 ± 0.17 ^a^	2.24 ± 0.02 ^a^
V48	2.79 ± 0.06 ^abc^	2.61 ± 0.03 ^a^	2.98 ± 0.01 ^bc^
Superior	3.37 ± 0.03 ^cd^	7.01 ± 0.15 ^c^	6.75 ± 0.07 ^gh^
N109-35	2.63 ± 0.01 ^ab^	17.36 ± 0.26 ^j^	3.33 ± 0.10 ^c^
N2-6	5.58 ± 0.26 ^f^	10.27 ± 0.23 ^e^	8.70 ± 0.23 ^k^
A9	4.66 ± 0.10 ^e^	12.73 ± 0.30 ^g^	3.37 ± 0.14 ^c^
A165-4	9.86 ± 0.02 ^i^	9.96 ± 0.20 ^e^	7.24 ± 0.46 ^hij^
V17	3.13 ± 0.02 ^bcd^	8.99 ± 0.24^d^	2.52 ± 0.10 ^ab^
A12-5	5.15 ± 0.06 ^ef^	37.92 ± 0.12 ^o^	4.97 ± 0.03 ^d^
V63	2.36 ± 0.04 ^a^	11.19 ± 0.04 ^f^	2.81 ± 0.08 ^abc^
N357	2.42 ± 0.05 ^a^	6.94 ± 0.07 ^c^	2.32 ± 0.27 ^a^
V18	2.85 ± 0.10 ^abc^	21.96 ± 0.04 ^m^	6.93 ± 0.09 ^ghi^
N69-2	7.05 ± 0.10 ^g^	28.49 ± 0.31 ^n^	6.46 ± 0.09 ^fg^
N96-29	8.58 ± 0.27 ^h^	16.42 ± 0.10 ^i^	13.93 ± 0.23 ^m^
A35-6	6.65 ± 0.00 ^g^	12.41 ± 0.08 ^g^	6.07 ± 0.04 ^ef^
Gogu	3.71 ± 0.10 ^d^	18.71 ± 0.08 ^k^	9.68 ± 0.12 ^l^
N189	2.65 ± 0.44 ^ab^	15.40 ± 0.24 ^h^	4.61 ± 0.01 ^d^
A186-23	12.82 ± 0.28 ^k^	28.75 ± 0.34 ^n^	19.18 ± 0.04 ^n^
A188-10	11.36 ± 0.00 ^j^	37.87 ± 0.00 ^o^	7.47 ± 0.10 ^ij^
V16	2.82 ± 0.02 ^abc^	40.84 ± 0.47 ^p^	5.78 ± 0.2 7^e^
A27	5.70 ± 0.03 ^f^	21.11 ± 0.06 ^l^	7.62 ± 0.22 ^j^
**Mean**	**4.96 ± 0.09**	**16.52 ± 0.16**	**6.11 ± 0.11**

Table results expressed as mean ± standard deviation. The means with different letters in each column are significantly different with *p* < 0.05 in Tukey’s test. The Bold is used to contrast the average mean against individual values

**Table 3 foods-08-00098-t003:** Reducing sugar levels after cold storage and after reconditioning.

Reducing Sugar (mg·g^−1^ DW)			
Clones	Cold Storage (4 °C)	Reconditioned (22 °C)	Recondition Ratio (%)
V50	7.02 ± 0.10 ^c^	1.73 ± 0.06 ^a^	75.4
V93	3.43 ± 0.04 ^b^	1.85 ± 0.03 ^a^	46.1
Dubaek	2.56 ± 0.17 ^a^	1.92 ± 0.08 ^a^	25.0
V48	2.61 ± 0.03 ^a^	2.18 ± 0.01 ^ab^	16.7
Superior	7.01 ± 0.15 ^c^	2.23 ± 0.13 ^ab^	68.2
N109-35	17.36 ± 0.26 ^j^	2.42 ± 0.00 ^abc^	86.1
N2-6	10.27 ± 0.23 ^e^	2.43 ± 0.15 ^abc^	76.3
A9	12.73 ± 0.30 ^g^	2.72 ± 0.03 ^abc^	78.6
A165-4	9.96 ± 0.20 ^e^	3.34 ± 0.01 ^abc^	66.4
V17	8.99 ± 0.24 ^d^	3.58 ± 0.08 ^abc^	60.2
A12-5	37.92 ± 0.12 ^o^	3.94 ± 0.04 ^bc^	89.6
V63	11.19 ± 0.04 ^f^	4.22 ± 0.01 ^cd^	62.3
N357	6.94 ± 0.07 ^c^	6.05 ± 0.09 ^de^	12.8
V18	21.96 ± 0.04 ^m^	6.60 ± 0.11 ^e^	69.9
N69-2	28.49 ± 0.31 ^n^	6.96 ± 0.06 ^e^	75.6
N96-29	16.42 ± 0.10 ^i^	7.27 ± 0.06 ^e^	55.8
A35-6	12.41 ± 0.08 ^g^	7.38 ± 0.24 ^e^	40.5
Gogu	18.71 ± 0.08 ^k^	13.99 ± 0.16 ^f^	25.2
N189	15.40 ± 0.24 ^h^	15.05 ± 0.24 ^f^	2.2
A186-23	28.75 ± 0.34 ^n^	17.01 ± 2.11 ^g^	40.8
A188-10	37.87 ± 0.00 ^o^	18.45 ± 0.13 ^g^	51.3
V16	40.84 ± 0.47 ^p^	20.53 ± 0.23 ^h^	49.7
A27	21.11 ± 0.06 ^l^	21.97 ± 0.74 ^h^	−4.0

Table results expressed as mean ± standard deviation. The means with different letters in each column are significantly different with *p* < 0.05 in Tukey’s test.

**Table 4 foods-08-00098-t004:** Potato chip color scores after frying tests in relation to different storage conditions.

Snack Food Association (SFA) Chip Color measurement ^z^			
Clones	Cold Storage (4 °C)	Cold Storage (10 °C)	Reconditioned (22 °C)
A9	2.0	2.5	2.5
A188-10	4.5	4.0	4.0
V16	3.0	3.5	4.0
A12-5	4.0	3.5	3.0
V17	3.0	2.5	2.5
V93	1.0	2.0	1.5
N357	3.0	1.5	2.5
V48	2.5	1.5	1.5
Superior	2.5	2.5	1.5
Dubaek	1.5	1.5	1.0
V63	3.0	3.0	3.0
V18	3.5	3.5	3.5
A165-4	2.5	4.5	2.0
V50	3.0	1.5	1.5
N2-6	3.0	2.5	2.0
N189	3.5	3.0	3.5
N69-2	3.0	3.0	3.0
N96-29	4.0	3.5	3.5
A27	3.5	3.5	4.5
Gogu	4.5	4.5	4.5
N109-35	3.0	1.5	3.5
A35-6	3.0	3.0	3.0
A186-23	4.5	5.0	4.5

^Z^ SFA scores measured on a scale of 1–5, with 1 = light color, and 5 = dark color.

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
