# Peer review of "Selection and Evaluation of 21 Potato (Solanum Tuberosum) Breeding Clones for Cold Chip Processing"

_foods, 2019, doi:10.3390/foods8030098_

Reviewer 1 Report

Point #1:- 

Abstract:- The Author stated that there are 4 clones with superior quality while it was stated in the Conclusion (Line # 229) that they were THREE clones only. The Author needs to be constant and that the results in agreement. 

Point #2:- 

INTRODUCTION needs support. The Author needs to include all the parameters with background and importance in the introduction to cover all the study points

Point #3:-

Some Typos and error for example Line # 85 and blot died---- should be blot dried. Just double check for any typo or error in the manuscript

Point #4:-

Statistical analysis in this kind of study is very critical . The Author needs to do statistical for all the measurements. in order to draw an accurate conclusion. For example in Table #4. The stats is very important . In this parameters a large # of chips should be evaluated and the stats will give a clear conclusion.

Point #5:-  

Results and Discussion Table:3 Reducing sugar level after cold----

The data in the table showed 4 significant and outstanding clones and yet it was stated that they were 8. Please explain and provide justification. 

Point #6:-  

Also in Table 3

RECONDITION RATIO???

PLEASE EXPLAIN HOW IT WAS CALCULATED AND PROVIDE The MEAN COMPARISON OR ANY KIND OF STATS for these data

Point #7:- 

Line # 200 The study observation is not clear

Point #8:- 

In page.6 Line # 214-215 The total glycoalkaloids content considered safe for consumption should not exceed 20 mg/100 gm FW ( WHERE IS THE REFERENCE )???

Point #9:- 

The pictures quality for the chips was average it could be more improved to provide the differences in color between the clones

Point #10:- 

In general Data presentation could be improved by using more graphs than tables to show the differences between the clones

Author Response

Review 1 comments

Point 1: Abstract: - The Author stated that there are 4 clones with superior quality while it was stated in the Conclusion (Line # 229) that they were THREE clones only. The Author needs to be constant and that the results in agreement.

Response 1: The 4clones were selected based on dry matter content and specific gravity only but the final 3clones (line 229) based on overall experimental results. For consistency purposes, we have included only the final 3 clones that performed exceptionally well.

Point 2: INTRODUCTION needs support. The Author needs to include all the parameters with background and importance in the introduction

Response 2: Importance and relevance of Tuber dry matter content (lines 126-129)

Importance of Tuber specific gravity (132-133)

Importance of reducing sugar (160-164)

Relevance and importance of Glycoalkaloids (219-221)

All parameters have been discussed and justified in the discussion section.

Point 3: Some Typos and error for example Line # 85 and blot died---- should be blot dried. Just double check for any typo or error in the manuscript

Thanks for pointing this out, we have checked the papers for such errors

Point 4: Statistical analysis in this kind of study is very critical . The Author needs to do statistical for all the measurements. in order to draw an accurate conclusion. For example in Table #4. The stats is very important . In this parameters a large # of chips should be evaluated and the stats will give a clear conclusion.

Response 4: Table 4 was based on observation and reading of the SFA table for the fried color chips based on the SFA color codes. Readings were recorded directly as read according to the SFA chart.

Line 239

Point 5: Results and Discussion Table:3 Reducing sugar level after cold----

The data in the table showed 4 significant and outstanding clones and yet it was stated that they were 8. Please explain and provide justification. 

Response 5: As per explanation in line 180-183…

Based on the reconditioning trends, the study observed although some clones showed a great reconditioning ability and good color chip their reducing sugar levels were still very high. This is a confirmation that, the degree to which quality can be restored with reconditioning varies with cultivar

Point 6: Also in Table 3

RECONDITION RATIO???

PLEASE EXPLAIN HOW IT WAS CALCULATED AND PROVIDE The MEAN COMPARISON OR ANY KIND OF STATS for these data

Ratio was basically calculated as a percentage of reconditioned tubers against non- reconditioned ones

Line 177: A genotype with safe reconditioning ratio would be desirable for the chip processing industry as it exhibits consistencies in reducing sugar levels both in cold storage and in reconditioned states.

#182 100 – [Reconditioned state/ cold storage state x 100]

Point 7: Line # 200 The study observation is not clear

Response 7: line 200 is figure 1 showing the top 10 pictures of fried chip samples from cold storage as compared to the set standard od dubaek and superior.

Point 8: In page.6 Line # 214-215 The total glycoalkaloids content considered safe for consumption should not exceed 20 mg/100 gm FW ( WHERE IS THE REFERENCE )???

Response 8:  we have included the reference [20]

Point 9: The pictures quality for the chips was average it could be more improved to provide the differences in color between the clones

Response 9: The clones had very slight color variations and we did our best to capture the initial colors immediately after the chip frying test.

We used a high resolution Canon EOS 5D Digital SLR Camera to take the best quality possible. We have now included this information in the study. (line 87-88)

Point 10: In general Data presentation could be improved by using more graphs than tables to show the differences between the clones

Response 10: We had tried presenting the results in graphs format but we felt it was not appropriate for this study, except for the glycoalkaloid contents.

Presenting 23 clones of potatoes in a single or double graph axes makes it quite complex and difficult to view at a glance as compared to the tables.

Reviewer 2 Report

M&M section. 

2.5 Chips frying test. 

For a better clarification, please add the required information:

-Could the authors specify and report the method used to photograph the chips?

-Could the authors describe the color measure method used and add some reference, based on the SFA

Author Response

Review 2 comments

Point 1: 2.5 Chips frying test. 

For a better clarification, please add the required information:

Reconditioning Ratio was basically calculated as a percentage of reconditioned tubers against non- reconditioned ones

Line 177: A genotype with safe reconditioning ratio would be desirable for the chip processing industry as it exhibits consistencies in reducing sugar levels both in cold storage and in reconditioned states.

100 – [Reconditioned state/ non-reconditioned state x 100]. We have added the calculation formulas and chip SFA colors.

Point 2: Could the authors specify and report the method used to photograph the chips?

Response 2: We used a high resolution Canon EOS 5D Digital SLR Camera to take the best quality possible. We have now included this information in the study. (line 87-88)

Point 3: Could the authors describe the color measure method used and add some reference, based on the SFA

Response 3: line 239 We have included the SFA color chart used in evaluating the potato chips

Round  2

Reviewer 1 Report

The Author provides the correction needed, Therefore the manuscript is ready for submission.

One comment Figure S1. Snack Food Association (SFA) chip color measurement standard

I strongly suggest to add this figure to the original manuscript.

Author Response

Thank you for your review comments, indeed we have included the SFA image (Figure 1) in the main manuscript text as per your recommendation.